# Using a Two-Sex Life Table Tool to Calculate the Fitness of *Orius strigicollis* as a Predator of *Pectinophora gossypiella*

**DOI:** 10.3390/insects11050275

**Published:** 2020-04-30

**Authors:** Shahzaib Ali, Sizhe Li, Waqar Jaleel, Muhammad Musa Khan, Jintao Wang, Xingmiao Zhou

**Affiliations:** 1Hubei Insect Resources Utilization and Sustainable Pest Management Key Laboratory, College of Plant Science and Technology, Huazhong Agricultural University, Wuhan 430070, China; shahzaibali@webmail.hzau.edu.cn (S.A.); 15927032739@sina.cn (S.L.); 2Key Laboratory of Bio-Pesticide Innovation and Application, Guangzhou 510640, China; Waqar4me@yahoo.com; 3Engineering Research Center of Biological Control, Ministry of Education, Guangzhou 510642, China; 4Department of Entomology, College of Agriculture, South China Agricultural University, Guangzhou 510642, China; 5Plant Protection Research Institute, Guangdong Academy of Agricultural Sciences, Guangzhou 510640, China; 6Key Laboratory of Bio-Pesticide Innovation and Application, Engineering Research Centre of Biological Control, South China Agricultural University, Guangzhou 510642, China; drmusakhan@outlook.com; 7Key Laboratory of Integrated Pest Management on Crops in Central China, Ministry of Agriculture, Institute of Plant Protection and Soil Fertility, Hubei Academy of Agricultural Sciences, Wuhan 430064, China; wjt217@126.com

**Keywords:** age-stage, feeding potential, *Orius strigicollis* Poppius, *Pectinophora gossypiella*, population parameters, selection pressure, two-sex life table

## Abstract

A two-sex life table is a useful tool for studying the fitness of predators. Previous studies of *Orius*
*strigicollis* Poppius (Heteroptera: Anthocoridae) fitness have not been done on *Pectinophora gossypiella* (Lepidoptera: Gelechiidae) using a two-sex life table tool. This study reports the fitness of the minute predatory flower bug, *O. strigicollis* when feeding on the cotton pest *P. gossypiella* using a two-sex life table tool. Different densities (5, 10, and 15 eggs) of *P. gossypiella* eggs were used to calculate the feeding capacity and fitness of *O. strigicollis* in the laboratory at 28 °C ± 1, 75 ± 5% RH and 16:8 (L:D). The results concluded that *O. strigicollis* is an efficient predator of *P. gossypiella*. The maximum growth capacity of the predatory bug *O. strigicollis* was attained when it was fed on 10 and 15 *P. gossypiella* eggs. Furthermore, shorter generation and development time were also observed in the case of 15 eggs of *P. gossypiella*. These results suggest that *O. strigicollis* has considerable predatory potential and prefers feeding on *P. gossypiella* eggs than on the first instar larvae at the fourth instar or the female stage. Although the field potential of *O. strigicollis* is still unknown, this study will support future investigations in terms of field applications.

## 1. Introduction

Natural enemies of insect pests feed on different insects belonging to different families [1,2]. The potential of a predator usually depends on its voracity (wanting or devouring great quantities of prey or food), area of discovery, functional response to prey density, mutual interference, reproductive, behavioral and numerical responses and spatial heterogeneity. Prey density is one of the important factors contributing to the competition or interference between the predator and prey [3,4]. The nutritional specificity creates a challenge for classical biological control programs, including those for predators and the predator and prey interactions, is an overwhelming challenge. Furthermore, the fitness of natural enemies is key to the success of efficient biological control systems, worldwide [5,6,7,8,9,10]. However, there are many factors that affect the fitness traits of predators such as prey quality, density and temperature [10,11]. Voracious or keen prey is also an important factor [11,12,13,14] that affects the development or fitness of predators [9,15,16,17,18,19,20,21].

*Pectinophora gossypiella* (Lepidoptera: Gelechiidae), is a cotton pest that decreases yields by up to 20–67% worldwide [22,23]. This monophagous pest continuously causes serious damage to cotton crops, and has been known to cause 2.8% to 61.9% losses in seed cotton yields, 2.1% to 47.1% losses in oil content, and 10.7% to 59.2% losses in the normal opening of bolls in China [24]. Controlling *P. gossypiella* populations through the application of different chemicals is difficult owing to their concealed modes of feeding [25], and the development of insecticide resistance in various strains of *P. gossypiella* in the previous decades [26,27,28,29]. The safest and most eco-friendly option for reducing the populations of *P. gossypiella* is using biocontrol agents, e.g., predators [8,30]. 

The best predator must have the ability to survive on availability of the prey [31]. There is, however, a high level of selection pressure on predators for prey quality and density, suitable environments for oviposition, and availability of antagonists [12,13,32,33,34]. Therefore, in complex ecosystems, each species intermingles with other species in many ways such as predatory, parasitic or mutualistic and competitive [35]. Among all interactions of species, predator–prey interaction is very important as it helps in suppressing insect pest population [36,37]. It is essential to evaluate the biocontrol potential of predators, e.g., *Orius* spp., in the laboratory by conducting biological studies. Moreover, these studies can justify the efficiency of predator against target species before using predators in the field against insect pests [38,39], and helpful in the regulation of prey population [12,40].

In biological control system, e.g., *Orius* spp (Heteroptera: Anthocoridae), are the most effective predators of lepidopteran [7,41], hemipteran, arachnids and various arthropods pests, i.e., bollworms (Lepidoptera: Noctuidae), whiteflies (Hemiptera: Aleyrodidae), aphids (Hemiptera: Aphididae) [42,43], mites (Arachnida: Acaridae), and other arthropod eggs [7,32,44]. Among *Orius* spp., the *Orius strigicollis* Poppius (Heteroptera: Anthocoridae), which was previously known as *Orius similis* Zheng (Heteroptera: Anthocoridae) (junior synonym of *O. strigicollis*) [45,46], also called “minute predatory flower bug”, has been found to be the most effective predator of insect pests [30,33,42].

The fitness of predator evaluated by the life table tool. Nowadays, two-sex life table tools are very important source to evaluate the fitness of predator on prey. Two-sex life table studies enable us to better understand the ecology and fitness of an organism, as they take both sexes into consideration to construct the correct population curve for the upcoming populations [47,48]. These studies are thus pivotal for determining the fitness and efficacy of predators, to understand the predator–prey relationship. Several studies have reported the fitness of *Orius* spp. with different prey species [49,50,51,52,53]. Moreover, some studies on the feeding preferences of *O. strigicollis* for different types of prey revealed that development, survival, and efficiency of the predator depends on the availability of the food source and suitable environmental conditions [6,7,30,54]. However, to the best of our knowledge, no comprehensive information regarding the fitness of the predatory bug *O. strigicollis* on *P. gossypiella*, using age-stage, two-sex life tools, has been reported previously.

This study addresses by examining the interactions between *O. strigicollis* and *P. gossypiella.* The objectives of this study were to determine (1) the feeding potential and efficacy of *O. strigicollis* under the selection pressures of different prey densities; (2) the fitness traits and population parameters of *O. strigicollis*; (3) prey preferences among *P. gossypiella* eggs and first instar larvae. The results of this investigation could help to maintain the predatory agent *O. strigicollis* against *P. gossypiella*, by considering the importance of Integrated pest management (IPM).

## 2. Materials and Methods

### 2.1. Insect Cultures

#### 2.1.1. Rearing of *O. strigicollis*

The *O. strigicollis* population was collected from different cotton fields of Huazhong Agricultural University, Wuhan (30.50° N latitude and 114.30° E longitude), Hubei Province, P.R. China. Cultures were confirmed to be of a single species by using the method previously described by [55], and the physical properties previously cited in the literature [45,46]. The *O. strigicollis* population was maintained on eggs of the pest *P. gossypiella* in the laboratory, using previously described techniques [30]. Stock cultures of *O. strigicollis* were maintained on surplus quantities of the *P. gossypiella* eggs for two generations at 28 °C for acclimatization to the prey and temperature, using previously described methods [21] before experimentation.

Small and soft stems (3–4) of *Vitex negundo* L. (Lamiaceae: Verbenaceae), were provided in plastic containers (100 mm wide at base and 124 mm deep), to *O. strigicollis* for oviposition/eggs laying under controlled conditions in a climatic chamber (HP250GS, Ruihua Instrument and Equipment Co., Ltd., Wuhan, China). Stems were covered with moistened cotton at the end and this was refreshed after two days. Stems containing *O. strigicollis eggs* were collected daily and kept in separate containers until the emergence of the adult. To avoid cannibalistic behaviors, every jar had small pieces of white foam as shelters and a sufficient number of eggs.

#### 2.1.2. Collection of *P. gossypiella*

The pupae and adults of *P. gossypiella* (resistant strain AZP-R, cotton) were provided by the Institute of Plant Protection and Soil Fertility, Hubei Academy of Agricultural Sciences, Wuhan, China, where worms have been continuously reared for over 50 generations on artificial diet. This diet was prepared by mixing the Wheat germ meal, Casein, Agar, Sucrose, Brewer’s yeast, α-cellulose, Potassium sorbate, Nipalgin, Decavitamin, Choline chloride, Maize oil, Honey, Water, Calcium pantothenate, Niacin, Riboflavin, Folic acid, Thiamine, Pyridoxine hydrochloride, Biotin and Vitamin B_12_ efficiently. Adults were placed in cages (50 × 50 × 50 cm) for oviposition and provided with 10% honey solution as an enhanced egg laying diet for the adults. Each cage was enclosed with white gauze and filter paper for oviposition. Egg clusters were collected daily on filter paper during the total oviposition period as previously described [56], and the fresh eggs were provided as a food source for *O. strigicollis*. All experiments were conducted at 28 °C, 75 ± 5% RH and 16:8 (L:D) photoperiods maintained in controlled chambers equipped with fluorescent lighting controlled by an automatic timer.

### 2.2. Feeding Potential of O. strigicollis

The feeding potential of *O. strigicollis* at different predatory stages (third, fourth, fifth nymphal instar, and adult stages (female and male)), was recorded on *P. gossypiella* eggs and the first instar larvae. Based on the preliminary experiments, three different densities (5, 10, and 15), of the *P. gossypiella* eggs and larvae as prey were selected for use in the main experiments. The feeding efficiency of all the predatory stages with the *P. gossypiella* eggs after 24 h, and for the first instar larvae after 12 and 24 h, were determined in plastic petri dishes (9 cm in diameter and 2 cm in depth), lined with filter paper. There were thirty replicates for each treatment and each predatory stage. Each predatory stage was introduced as one individual with respective treatment in main experiment. We noted that, all the newly laid and healthy *P. gossypiella* eggs at the three different prey densities (5, 10, and 15) treatments, showed 100% hatchability without predator and we did not take them into account while analyzing the data. Therefore, no control treatments were used for the *O. strigicollis* potential on *P. gossypiella* eggs. Although thirty replicates of the control treatment for the *O. strigicollis* potentials on the *P. gossypiella* first instar larvae were used without the predator at the three different prey densities, for 12 and 24 h durations. The control treatment was a prediction of the *P. gossypiella* first instar larvae mortality, with and without the predator, to check the actual feeding potentials of *O. strigicollis* in the predatory stages on the *P. gossypiella* first instar larvae, by following the methodology of [32].

The total feeding of *O. strigicollis* all stages in the different prey densities when fed on the *P. gossypiella* eggs (5, 10, and 15) were determined during entire developmental stages. The first instar *O. strigicollis* nymphs (30) were selected and fed at each treatment until death.

### 2.3. Selection Pressure and Biological Parameters

The fitness and population, or life table parameters, were determined with the help of the age-stage, two-sex life table tool that efficiently explains the aspects of the prey range on the fitness and development of the predators [11,14,57,58]. Selection pressure is a barrier for the decision of suitable, favorable, or reliable sources of food for insects to complete their entire development or life stages, and the survival of future generations [11,12,13,14]. In this experiment, the selection pressure was provided to the predatory stages of *O. strigicollis* in the form of limited or excessive food (5, 10, and 15 eggs), and the alternate effects of this pressure on the entire generation was determined.

#### 2.3.1. Effects of Different Preying Densities on Developmental Period of *O. strigicollis*

The total number of freshly hatched *O. strigicollis* eggs (60, less than 12 h old), were selected and checked for the development time from the hatching of the egg to the second instar nymph (N_2_) in the form of groups (≤10 in each box) when fresh *P. gossypiella* eggs (5, 10, and 15) were provided (mentioned above by cutting filter paper into pieces), as prey separately. Small pieces of white foam as shelters to avoid cannibalism, and cotton balls drenched with water to maintain moisture levels and as a source of water for the immature nymphs, were provided. Starting with the third instar nymphal stage (N_3_), the individual bugs were secluded in plastic petri dishes, lined with filter paper to avoid cannibalism, as described by Tuan, et al. [59]. Each individual was considered a replicate at each preying density until death, to check the selection pressures on *O. strigicollis*. Each nymph was fed with the respective *P. gossypiella* egg densities, as diet/prey until the emergence of the adult, to check the predation capacity. The stadial duration of each stage was noted after every 24 h. Fresh *P. gossypiella* eggs were provided on daily basis and every 24 h the consumed and damaged eggs were counted with the help of a stereomicroscope (1309 LED 40X Binocular Stereo Microscope, Jiangsu Victor Instrument Meter Co., Ltd., Taizhou, China), and magnifying lens. The mortality of *O. strigicollis* was determined daily.

The moultings or shredded skins of the *O. strigicollis* nymphs in the round plastic petri dishes were washed and sterilized daily at each age for assessment of the next nymphal stage, by following the methodology of Zhang et al. [30]. The males and females were identified at the fifth nymphal stage with the stereomicroscope, following the methodology of Amer et al. [33]. After the emergence of the adults, the males and females were immediately shifted into a plastic box and starved for 24 h for the reproductive study.

#### 2.3.2. Longevity, Oviposition, and Fecundity of *O. strigicollis*

For adults, the longevity and egg laying capacity of *O. strigicollis*, the females and males were shifted to new cylindrical glassy vials (2.6 cm in diameter and 15 cm in length), enclosed with a fine mesh nylon screen. The pairs were noted, to ensure that mating occurred and the females that continued copulation for >1.5 min were supposed to have been mated, according to Butler [43], and no prey was added at this time to stimulate the mating process [60]. The pairs in each treatment (5, 10, and 15 eggs) were placed in the petri dishes, lined with filter paper and *P. gossypiella* eggs (10, 20, and 30 eggs/pair) were provided as prey for the adult pairings (double prey density). The small fragile stems of the *Vitex negundo* L. were provided as a substrate for the female *O. strigicollis* for oviposition, according to the method described by Zhou et al. [42], for each treatment (5, 10, and 15 eggs of *P. gossypiella*). The stems were covered with moistened cotton, as described above, and investigated daily using a stereomicroscope to count the number of eggs laid by *O. strigicollis*. The adults pre-oviposition period (APOP = the time period between the female adult emergence to its first egg laying), oviposition duration (d), and total pre-oviposition period (TPOP = the time interval between birth to the start of egg laying), were recorded until the time of death of the female predators. The consumed or damaged eggs of the *P. gossypiella* were counted and replaced with fresh eggs daily. The stems with eggs laid by the *O. strigicollis* female were kept in plastic containers and were placed in a chamber as described above, to check their biological or fitness parameters after each treatment following previously described methods [30,61,62,63]. These parameters were greatly influenced by the selection pressures on the predators, to continue the next generation or suppress the pest population dynamics successfully [10,11,12,13,59,64]. The number of *O. strigicollis* hatched eggs were recorded daily using a stereomicroscope. The percent of egg hatchability was determined as “the hatched eggs laid by one pair/total no of eggs laid by one pair × 100”. All the *O. strigicollis* bugs (female) were kept and observed until their death as described in previous investigations [30,33].

### 2.4. Prey Preference

*O. strigicollis* at different predatory stages (third, fourth, and fifth nymphal instar, and adult stages (female and male)), were exposed to the *P. gossypiella* eggs and the first instar larvae (10 each) in the same plastic petri dishes. There were ten replicates for each predatory stage. Each replicate was provided with moistened cotton placed inside the petri dish to maintain the moisture and water source for all predatory stages in the above-mentioned controlled chamber. The preferred or consumed prey were examined after 8-, 16-, and 24-h treatments under a stereomicroscope.

### 2.5. Life Table Analysis

In this study, according to completely randomized design (CRD), the data of the feeding potentials (including average prey killed and total feeding) of *O. strigicollis* were statistically analyzed using one-way ANOVA (analysis of variance) and their mean values were compared using least significant difference (LSD) tests at the *p* = 0.05 level of significance. All statistical analyses were performed using statistics 8.1 software (Analytical Software, Tallahassee, FL, USA). Different biological/fitness parameters (each stage developmental period, survival rate, adult longevity, age-specific fecundity, APOP and TPOP), were statistically evaluated using an age-stage, two-sex life table tool [65], and the TWOSEX-MS Chart computer program [58]. Means and standard errors (SE) of all the biological and population or life table parameters were determined using 200,000 bootstrap replicates to obtain stable estimates (SE) [47]. The bootstrap and paired bootstrap tests were determined in TWOSEX-MS Chart and the results of the treatments were run using the paired bootstrap test based on the confidence interval of difference [66]. The age-stage-specific survival rate (*s_xj_*: the possibility that newly laid eggs will live or exist to age *x* and stage *j*), age-stage-specific fecundity (*f_xj_*: the mean fecundity of females at age *x*), age-specific survival rate (*l_x_*: the possibility that newly laid egg will live or exist to age *x*), and age-specific fecundity (*m_x_*: the mean fecundity of individuals at age *x*), were designed in sequence according to Chi [67]. Sigma Plot 12.0 was used to make the curves for all demographic parameters.

In the age-stage, two-sex life table, *l_x_* and *m_x_* were calculated as: (1)$$l_{x}=\sum_{j=1}^{k}s_{xj}$$
(2)$$m_{x}=\frac{\sum_{j=1}^{k}s_{xj}f_{xj}}{\sum_{j=1}^{k}s_{xj}}$$
where *k* is the last stage of the study cohort.

The intrinsic rate of increase (*r*) was then predicted iteratively following the Euler–Lotka equation with age indexed from 0 as follows [68]:(3)$$\sum_{x=0}^{\infty }e^{-r\left(x+1\right)}l_{x}m_{x}=1$$

The net reproductive rate (*R*_0_) represents the total number of offspring that an individual can produce during its lifetime and is calculated as:(4)$$R_{0}=\sum_{x=0}^{\infty }l_{x}m_{x}$$

The net reproductive rate (*R*_0_) and mean female fecundity (*F*) relationship is as follows:(5)$$R_{0}=F\frac{N_{f}}{N}$$
where *N* indicates the total number of individuals used for the life table study and *N_ƒ_* represents the number of female adults [65].

The gross reproduction rate of individuals/insects is an indication of a rapid increase of insect populations that depends on the fecundity and adult eclosion, calculated as follows:(6)$$GRR=\sum_{x=0}^{\infty }m_{x}$$

The finite rate (*λ*) is recorded as:(7)$$\lambda =e^{r}$$

The mean generation time (*T*) represents the time span that a population needs to increase to *R*_0_-fold of its size, i.e., *e^rT^* = *R*_0_ or λ*^T^* = *R*_0_ at the stable age-stage distribution. The value of *T* is calculated as:(8)$$T=\frac{lnR_{0}}{r}$$

Age-stage life expectancy (*e_xj_*) is defined as, the length of the duration or time that an individual or insect of *x* and *j* is predicted to live, calculated by the method of Chi and Su [69] as:(9)$$e_{xj}=\overset{\infty }{\underset{i=x}{\sum }}\overset{k}{\underset{y=j}{\sum }}s_{iy}^{\prime }$$
where $s_{iy}^{\prime }$ is defined as the probability that individuals of *x* and *j* will survive to age *i* and stage *y*, and is found by assuming [48].
(10)$$s_{iy}^{\prime }=1$$

The age-stage reproductive value (*v_xj_*) was defined as the contribution of individuals of age *x* and stage *j* to the future population [70]. In the age-stage, two-sex life table, it is calculated as follows [48]:(11)$$v_{xj}=\frac{e^{-r\left(x+1\right)}}{S_{xj}}\;\overset{\infty }{\underset{i=x}{\sum }}e^{-r\;\left(x+1\right)}\overset{k}{\underset{y=j}{\sum }}s_{iy}^{\prime }f_{iy}$$

Prey preferences between the two prey types, i.e., the *P. gossypiella* eggs and the first instar larvae, were determined by using a paired *t*-test.

## 3. Results

### 3.1. Feeding Potential of O. strigicollis

Feeding potential of *O. strigicollis* predatory stages, i.e., third, fourth, fifth nymphal instar, adult stages (female and male), and control (no predator), were recorded after being fed the *P. gossypiella* eggs (Figure 1), and first instar larvae, as shown in Figure 2 and Figure 3. Among all predatory stages for the *P. gossypiella* eggs, the female showed significantly increased feeding capabilities (9.53 ± 0.22 eggs/female) with the high prey density of 15 eggs, compared to the 5 egg (3.87 ± 0.16 eggs/female) and 10 egg (7.57 ± 0.28 eggs/female) (F_2,87_ = 161; *p* < 0.01) densities, after 24 h. Similarly, the male showed significantly increased feeding capabilities (9.27 ± 0.22 eggs/male) with the high prey density of 15 eggs, compared to the 5 egg (3.77 ± 0.14 eggs/male) and 10 egg (7.17 ± 0.33 eggs/male) (F_2,87_ = 133; *p* < 0.01) densities, after 24 h (Figure 1).

Similarly, all predatory stages with the *P. gossypiella* larvae showed different feeding capabilities with the high prey density after 12 and 24 h intervals. The females feeding levels significantly increased with the 15 larvae (10.13 ± 0.32 larvae/female), compared with the 5 larvae (3.23 ± 0.22 larvae/female) and 10 larvae (7.06 ± 0.14 larvae/female) densities after 12 h (F_2,87_ = 213; *p* < 0.01). The male feeding levels were also significantly increased with the 15 larvae (9.97 ± 0.31 larvae/male), compared with the 5 larvae (3.13 ± 0.20 larvae/male) and 10 larvae (7.17 ± 0.12 larvae/male) densities after 12 h (F_2,87_ = 241; *p* < 0.01) (Figure 2). Furthermore, the female fed significantly more with the 15 larvae (11.17 ± 0.15 larvae/female) treatments, compared with the 5 larvae (4.03 ± 0.18 larvae/female and 10 larvae (8.67 ± 0.22 larvae/female) treatments, after 24 h (F_2,87_ = 381; *p* < 0.01), and the males also fed significantly more in the 15 larvae (10.40 ± 0.25 larvae/male) treatments than the 5 larvae (3.97 ± 0.18 larvae/male) and 10 larvae (8.43 ± 0.21 larvae/male) treatments, after 24 h (F_2,87_ = 233; *p* < 0.01) (Figure 3). There were no differences recorded in the average consumptions of the males and females on the *P. gossypiella* first instar larvae after 12 and 24 h intervals. After 12 h of the control treatments (F_2,87_ = 20.2; *p* < 0.01), there was a significant difference between the male (F_2,87_ = 241; *p* < 0.01) and female (F_2,87_ = 213; *p* < 0.01), predatory stages (Figure 2). Similarly, after 24 h of the control treatments (F_2,87_ = 17.5; *p* < 0.01), there was a significant difference between the third instar (F_2,87_ = 122; *p* < 0.01), and fourth instar (F_2,87_ = 157; *p* < 0.01) predatory stages (Figure 3).

The total feeding of *O. strigicollis* in the different prey densities when fed on the *P. gossypiella* eggs, are presented in Table 1. First and second instar bugs fed fewer prey eggs over their entire lives, because at the initial immature stages, they were mostly fed on plant tissues and the moisture content that was provided by the petri dishes. The third instar nymphs fed significantly more eggs (*F*_2,79_ = 9.43; *p* < 0.01) when fed 15 eggs, compared with the 5- and 10-egg treatments, during the entire nymphal development period. Similarly, the fourth instar nymphs fed significantly more eggs (*F*_2,66_ = 7.44; *p* < 0.01) when fed 15 eggs, compared with the 5- and 10-egg treatments, during their entire nymphal development. Moreover, there were significant differences recorded between the female feedings during the entire development period, until death (Table 1).

### 3.2. Biological Parameters of O. strigicollis

#### 3.2.1. Developmental Period

The development time (days) of the egg to the pre-adult stages was determined at each prey density of the *P. gossypiella* eggs (5, 10, and 15 eggs) (Table 2). There were no differences recorded from the egg to the second instar nymph development period at each prey density (treatment). There were differences recorded in the mean development times at the third instar for the 15 eggs (2.31 ± 0.20 d), when compared with that of the 5 eggs (3.38 ± 0.12 d) and 10 eggs (3.00 ± 0.19 d), and at the fourth instar for the 15 eggs (1.93 ± 0.11 d), when compared with the 5 eggs (2.55 ± 0.11 d) and 10 eggs (2.07 ± 0.14 d) (Table 2).

#### 3.2.2. Longevity, Oviposition, and Fecundity of Adults

The total longevity of the adult males was highest with 10 eggs (29.50 ± 2.49 d), when compared with the 5 eggs (26.00 ± 2.09 d) and 15 eggs (23.67 ± 1.08 d), while the total longevity of the adult females was highest with the 10 eggs (29.20 ± 1.91 d), when compared with the 5 eggs (28.60 ± 1.73 d) and 15 eggs (26.12 ± 2.10 d) (Table 2). However, there were no differences recorded in the total longevity of the male and female adults among the three treatments. The TPOP of the adult females were different when fed 5 eggs (19.80 ± 0.13 d), 10 eggs (18.00 ± 0.36 d), and 15 eggs (17.50 ± 0.50 d) of the *P. gossypiella*. There were no differences recorded in the APOP of the adult females, but the shortest duration of the APOP was found to be with the 15 eggs (0.25 ± 0.11 d), when compared with the 5 eggs (0.40 ± 0.16 d) and 10 eggs (0.70 ± 0.17 d). Similarly, there were no differences recorded in the oviposition period for the three tested treatments (Table 3).

The highest fecundity (total no of eggs per female) was recorded with the 10 eggs (90.44 eggs/female), when compared with the 5 eggs (54.40 eggs/female) and 15 eggs (66.12 eggs/female) treatments. There were no differences recorded in the hatching percentage of the eggs laid by the *O. strigicollis* females in the three treatments (Table 3). Similarly, no differences were recorded in the mating pair success (MPS%) of *O. strigicollis* (the percentage of pairs of adults that successfully laid eggs or were eligible to continue to the next generation) with the 5 eggs (80%), 10 eggs (93.33%) and 15 eggs (93.33%).

#### 3.2.3. Population Parameters of *O. strigicollis*

The population or life table parameters, *r*, *λ*, *R*_0_, *T* and *GRR* of *O. strigicollis* that were fed the different prey densities of the *P. gossypiella* eggs, were calculated using the bootstrap technique with 200,000 resamplings (Table 4). The values of the *r* and *λ* of *O. strigicollis* that were fed on the *P. gossypiella* eggs were higher when fed on 15 eggs (0.14 ± 0.01 d^−1^ and 1.15 ± 0.02 d^−1^), compared with the 5 eggs (0.09 ± 0.02 d^−1^ and 1.10 ± 0.02 d^−1^), respectively. There were no differences observed between the *r* and *λ* after the three treatments. In addition, the value for *R*_0_ was highest with the 10 eggs (27.13 offspring), compared the other two treatments of 5 eggs (9.07 offspring) and 15 eggs (17.63 offspring). Whereas, *T* was higher and different with the 5 eggs (23.57 days), when compared with the 10 eggs (23.21 days) and 15 eggs (20.89 days). *GRR* was higher with the 10 eggs (129.61), when compared with the 5 eggs (51.03) and 15 eggs (90.32) (Table 4).

#### 3.2.4. Survival Rate

The detailed survival rate (*s_xj_*), of *O. strigicollis* that were fed on the three different densities of the *P. gossypiella* eggs are shown in Figure 4. The results showed the probability that a newly hatched predator will live to age *x* and stage *j*. Significant differences were observed in the overlapping plotted curves for the different developmental stages, because the developmental rate varied among the individuals at the different selection pressures of each prey density for each specific stage. The projected curves revealed the entirely diverse patterns for each developmental stage, for each treatment. The overall pre-adult development time was longer and the survival rate lower, when *O. strigicollis* were fed 5 eggs of the *P. gossypiella*. While the survival of the females and males was higher when 15 eggs were given, compared with 10 eggs (Figure 4).

The curves of *l_x_*, *m_x_*, *f_xj_* and *l_x_m_x_* that are plotted for the three different prey densities, are shown in Figure 5. The curves of the *l_x_* (basic form of *s_xj_*) showed a direct relationship or totally dependence on the prey density. The peak recorded values of the *f_xj_* were, 22 d (11.25 eggs), 16 d (18 eggs), and 17 d (18 eggs), appeared when fed 5, 10, and 15 eggs of the *P. gossypiella*, respectively. The curve of the age specific fecundity (*m_x_*), showed that the reproduction began at different ages with various prey densities, i.e., 18 days with 5 eggs, 15 days with 10 eggs, and 14 days with 15 eggs of the *P. gossypiella* (Figure 5).

#### 3.2.5. Life Expectancy

The effects of the three different treatments for prey-density on the predicted average life of the population (*e_xj_*), for every stage of *O. strigicollis* were plotted (Figure 6). The longevity of *O. strigicollis* from age zero was 16.48 days with 5 eggs, which differed from the 20.26 and 17.80 days with the 10 and 15 eggs of *P. gossypiella*, respectively. With the variation of the other developmental stages, an increasing trend of the especially female adult peak expectation was observed (16.5 d at age 23 d) with 15 eggs, compared with the 5 eggs (10 d at age 28 d) and on 10 eggs (12.69 d at age 16 d), and then gradually decreased (Figure 6).

#### 3.2.6. Reproductive Value

The age-stage reproductive value (*v_xj_*), describes the contribution of an individual of age *x* and stage *j* towards the upcoming population (i.e., the scale of the population forecasting). The curves for the reproductive values significantly increased when reproduction began, and the reproductive values were the same as the finite rate, i.e., 1.10 d^−1^ with 5 eggs, 1.15 d^−1^ with 10 eggs, and 1.15 d^−1^ with 15 eggs of *P. gossypiella*. The curve for the reproductive value was highest with the 10 and 15 eggs of the *P. gossypiella*, as compared with the 5 eggs. The *v_xj_* values for *O. strigicollis* in each treatment are shown in (Figure 7).

### 3.3. Prey Preference

The prey preference for the *P. gossypiella* eggs and the first instar larvae, was evaluated by using paired *t*-tests. Third instar nymphs remarkably preferred the *P. gossypiella* eggs than the first instar larvae at each treatment, i.e., 8 h (*t* = 3.28; *p* < 0.01), 16 h (*t* = 2.71; *p* < 0.05) and 24 h (*t* = 3.21; *p* < 0.05) intervals. Whereas, fourth instar and fifth instar preferred the *P. gossypiella* eggs than the first instar larvae only after 8 h (fourth instar; *t* = 3.25; *p* < 0.05, fifth instar; *t* = 3.21; *p* < 0.05) intervals than other treatments. Male and female preferences were not different but preferred the first instar larvae of the *P. gossypiella* than the other predatory stages (Figure 8).

## 4. Discussion

The feeding potential and fitness of predator play an important role in predator–prey relationship and is affected by many factors such as prey size and density [71,72], environmental factors, i.e., temperature [73,74], complexity of the habitat, and the internal state of the predator [11,21,75]. Among them, voracious prey is one of the most important factors [11,12,13,14], which contributes considerably to the biology or fitness, physiology, fecundity, longevity, behavior, survival, and feeding efficiency of predators [9,15,16,21]. To evaluate the fitness and potential of a predator to tackle agricultural pests, feeding or predating potential rates are an important constraint [33,34,59]. For these studies, two-sex life tables provided the basic and vital information for the ecological and biological approaches, as compared to the traditional life tables. Life table parameters are a useful method to assess the predator–prey relationship, population developments, survival, and reproduction [11,14,76]. Therefore, our study was designed, for the first time, to study predator–prey interactions by using the age-stage, two-sex life table tool to calculate the feeding potential and fitness traits of *O. strigicollis* under selection pressures when fed on different densities of the *P. gossypiella* eggs. In the present study, the life table tool explained that the maximum growth capacity or fitness of the predatory bug *O. strigicollis* could be attained when *O. strigicollis* was fed on 10 and 15 *P. gossypiella* eggs. The overall maximum population and fastest growth to avoid prey pressures could be achieved with 15 eggs, with short development and generation times. This study will help to improve IPM strategies against *P. gossypiella* by providing the required basic and additional information, such as the predating potential, fitness, and prey/food preferences of *O. strigicollis*. 

Previous investigations have shown the effects of different diets or preys on the survival and development of various insects [30,33,42,54,77,78]. Statistically significant distinctions in the pre-adult stages demonstrated that *O. strigicollis* were very receptive to changes in the prey density. Many studies have previously reported the phenomena that the nymphal developmental duration of *Orius* spp. was affected by different feeds or by the same prey species bearing strong selection pressures for successful development [5,32,33,79]. In the present study, the total nymphal duration was minimum when fed with a higher prey density as compared to when fed with a low prey density (Table 2). A conflicting relationship between longevity and food preference variation was observed in *Orius* spp. because suitable nutritious contents and ecological conditions fulfil the demands of insects at certain level, which could be attributed to the perfect diet or prey species [11,13,14,33,80]. Kakimoto et al. [81], revealed that the longevity of *O. similis* females and males was (27.2 d and 13.2 d) as compared to that of *O. sauteri* (23.8 d and 14.5 d) at 29 °C, when they were fed with *Ephestia kuehniella* Zeller. Aragon et al. [13] demonstrated that the total longevity of adults (males and females) declined when the adult *O. laevigatus* fed on *S. exigua* eggs (30.89 d) as compared to when they fed on *E. kuehniella* eggs (38.8 d). The same trend was observed in the present study with a little contrast, where the total longevity of the female and male adults of *O. strigicollis* was strongly influenced (increased) and changed owing to the prey/diet, except when *O. strigicollis* adults fed on 15 eggs of *P. gossypiella*, where the total longevity of females and males was lower (26.12 d and 23.67 d) as compared to when they fed on lower prey densities. Moreover, there was no significant difference between the three treatments (prey densities). This may be attributed to nutritional content or quality of the particular diet/prey, less food pressure to complete development, physical defense or response of the prey, more feeding resources [5,11,12,13,14,15,30,33,79,82,83,84], different preference behavior of the generalist predators to egg laying [5,7,16,42,78,85,86,87,88], and to complete metabolic reactions in the insect body under selection pressure [57,89,90]. We recorded that the females lived longer than the males (Table 2), which are in correlation with previous studies [30,32,33,42,44,54].

In the present study, the APOP of *O. strigicollis* was found to be prey density dependent. This correlates with the findings of [33], who reported that the pre-oviposition period of *O. similis* was the shortest on *C. cephalonica* (4.6 d) and longest on *A. craccivora* (8.3 d). However, in our study, the length of the APOP decreased on feedings 15 eggs of *P. gossypiella*, which showed that *O. strigicollis* could complete the development of the reproductive system (ovarioles) efficiently when *P. gossypiella* eggs were provided as a food, with respect to the low density or more pressure of food. The mean daily ovipositions of *O. similis* were different when fed two aphid species as prey; a maximum oviposition on *A. gossypii* (14.7 d) as compared to *M. persicae* (14.3 d), as documented by Ahmadi et al. [54]. Similarly, the study by Sengonca et al. [5] reported that the mean daily oviposition of *O. similis* reached a maximum on *M. persicae* (5.6 eggs/day) compared with that on *A. gossypii* (2.9 eggs/day) at 25 °C. This shows that the food also had an influence on the oviposition of the insects, according to the results of our study, where the oviposition period altered according to the diet density variations.

An early oviposition (egg laying) period under the selection pressures of a specific diet, may result in an increased metabolic rate, as reported in previous investigations [10,21,61,91]. The highest lifetime fecundity (overall oviposition per female) was observed with 10 eggs of *P. gossypiella* in our study (Table 3), which showed the egg laying potential of *O. strigicollis* with specific diets, in correlation to previous studies [14,17,18,30,33,49,54,81,92]. The *e_xj_* values of the eggs and females were lower and remarkably declined earlier for *O. strigicollis* that were fed 5 eggs compared with those fed 10 and 15 eggs of *P. gossypiella*, because of the increasing pressure of the prey (*P. gossypiella*). Meanwhile, the curves for the reproductive values (*v_xj_*) significantly increased when reproduction began, and the reproductive values were exactly the same as the finite rates (Table 4). In addition, when 5 eggs of the *P. gossypiella* were given to *O. strigicollis*, the expectancy (*e_xj_*) of the *O. strigicollis* eggs showed lower reproduction rates. This showed that *O. strigicollis* had greater reproduction potentials on the higher prey densities (10 and 15 eggs of *P. gossypiella*), enabling it to complete its development faster. Previous studies on the dipteran and lepidopteran insects have shown the same trends of results [11,61,76], that the reproduction rate and expectancy of the individuals was affected by the availability of the prey pressure and environmental factors, e.g., temperature.

For the growth, development, and survival of an insect, *r* is an important and critical demographic parameter [44,62,93,94]. The *r* is highly linked with the vulnerability of a prey or diet to insect feeding [95]. According to the demographic life table theory explained by the authors of studies [62,96], if *r* is greater than zero (*0*), then the prey (diet) is suitable for population growth. This theory supported the present study and surprisingly showed that *O. strigicollis* showed a higher and similar *r* (faster development and highest survival rates) when fed with 10 and 15 eggs of *P. gossypiella*, compared to the 5-egg treatment, owing to higher fecundity and shorter or faster development times (Table 4). In addition, our study was linked with other recent studies [44,81,94,97], whose authors reported the value of *r* for different *Orius* spp. and proved that diet has an important role in the approximating of the population parameters of insects.

The *R*_0_ is also a significant indicator of population development, where the highest rate of insects population is dependent and directly related to the number of eggs [98]. The *GRR* is considered a sign or concept of a rapidly increasing insect population that depends on fecundity and adult eclosion. Generally, these parameters are affected by food sources and temperature [9,21,63,99,100]. The dietary and thermal conditions for the selection of the prey have ultimate and direct impacts on the population parameters of the insect population [11,12,30,44,59,94]. In the present study, the highest net reproductive rates (*R*_0_) and gross reproductive rates (*GRR*) were achieved when the bugs were fed on 10 *P. gossypiella* eggs, when compared to those in the other treated groups (Table 4). These high growth rates must have been due to the quick development and high fecundity of *O. strigicollis*, because the quality of prey is directly related with the reproductive rates and developmental duration of the insects or predators [11,14,32,33,80]. The population increases only when the net reproductive rate (*R*_0_) is greater than one (*1*). Chi [65] showed the relationship between the net reproductive rate (*R*_0_) and mean female fecundity (*F*) as shown in equation 5, and our results were in accordance with this above-mentioned theory.

## 5. Conclusions

This study concluded that *O. strigicollis* is a potential predator for *P. gossypiella* eggs at different densities, but that the prey availability and preferences affected the growth parameters and population dynamics of the insects and predators. Our study suggests that the fitness of *O. strigicollis* is directly influenced by the predator densities. Under selection pressures of the different diet availabilities, *O. strigicollis* was an effective predatory agent and successfully completed its nymphal development, and the populations on the *P. gossypiella* eggs were confirmed using the age-stage, two-sex life table, which could be helpful for mass rearing of these bugs. Further studies are needed to determine the field efficacy of *O. strigicollis* against *P. gossypiella*.

## Figures and Tables

**Figure 1 insects-11-00275-f001:**
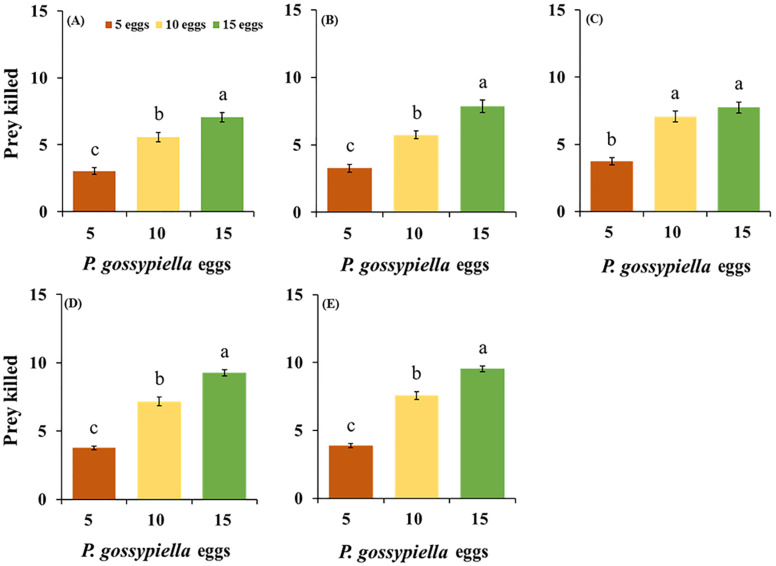
Feeding potential of the predatory stages of the *O. strigicollis* on three different densities of *P. gossypiella* eggs (5, 10 and 15) after 24 h, (**A**) = 3rd instar, (**B**) = 4th Instar, (**C**) = 5th Instar, (**D**) = Male, (**E**) = Female. Different letters above each bar indicate significant differences between three treatments using one-way ANOVA, LSD test, *p* = 0.05 and *n* = 30).

**Figure 2 insects-11-00275-f002:**
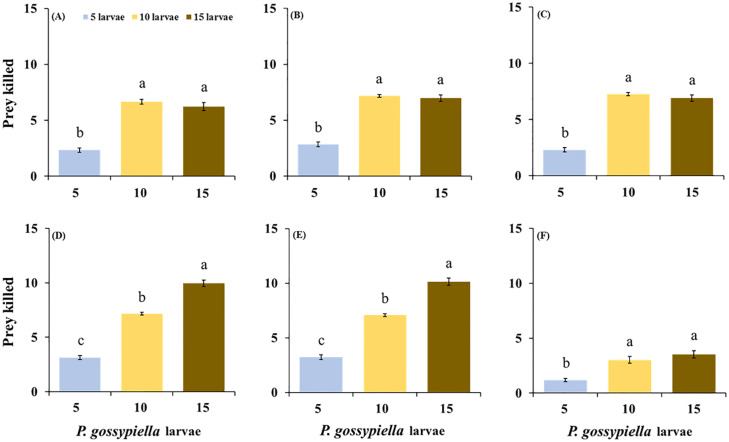
Feeding potential of the predatory stages of the *O. strigicollis* on three different densities of *P. gossypiella* first instar larvae (5, 10, and 15) after 12 h, (**A**) = 3rd instar, (**B**) = 4th Instar, (**C**) = 5th Instar, (**D**) = Male, (**E**) = Female, (**F**) = Control. Different letters above each bar indicate significant differences between three treatments using one-way ANOVA, LSD test, *p* = 0.05 and *n* = 30).

**Figure 3 insects-11-00275-f003:**
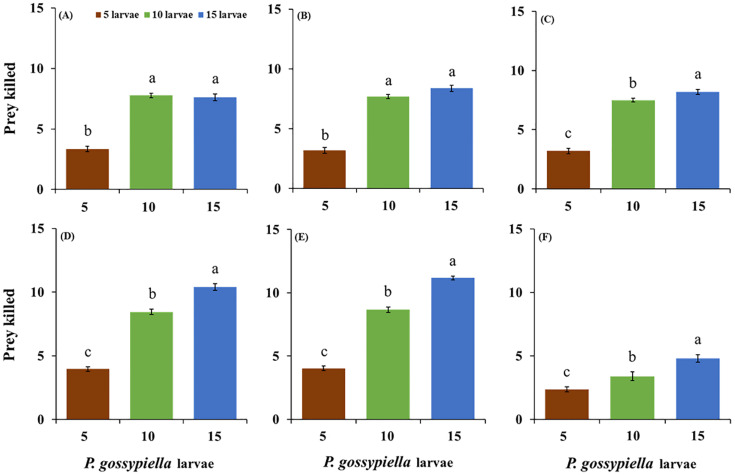
Feeding potential of the predatory stages of the *O. strigicollis* on three different densities of *P. gossypiella* first instar larvae (5, 10, and 15) after 24 h, (**A**) = 3rd instar, (**B**) = 4th Instar, (**C**) = 5th Instar, (**D**) = Male, (**E**) = Female, (**F**) = Control. Different letters above each bar indicate significant differences between three treatments using one-way ANOVA, LSD test, *p* = 0.05 and *n* = 30).

**Figure 4 insects-11-00275-f004:**
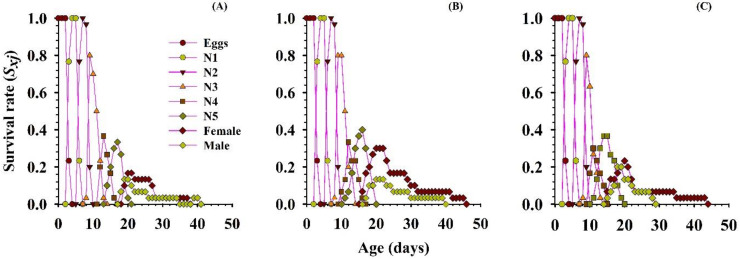
Influence of three different densities of *P. gossypiella* eggs ((**A**) = 5 eggs, (**B**) = 10 eggs and (**C**) = 15 eggs) on the age-stage-specific survival rate (*s_xj_*) of the *O. strigicollis*; N1 = 1st Instar, N2 = 2nd Instar, N3 = 3rd Instar, N4 = 4th Instar, N5 = 5th Instar.

**Figure 5 insects-11-00275-f005:**
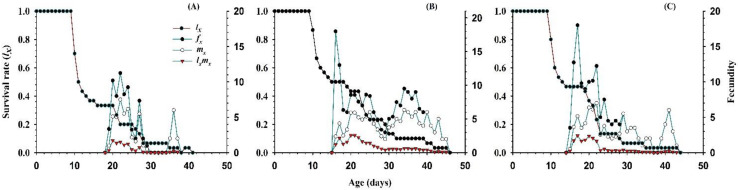
Influence of three different densities of *P. gossypiella* eggs ((**A**) = 5 eggs, (**B**) = 10 eggs and (**C**) = 15 eggs) on the age-specific survival rate (*l_x_*), female age-specific fecundity (*f_x_*), age-specific fecundity (*m_x_*), and age-specific maternity (*l_x_m_x_*) of the *O. strigicollis*.

**Figure 6 insects-11-00275-f006:**
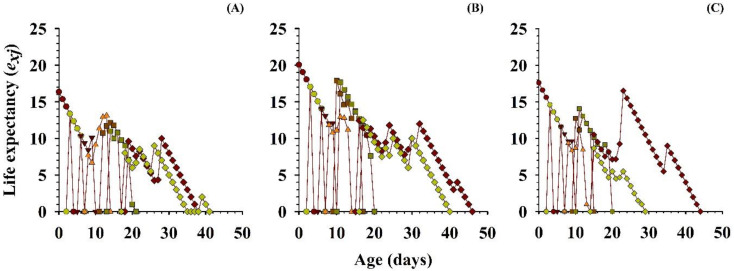
Influence of three different densities of *P. gossypiella* eggs ((**A**) = 5 eggs, (**B**) = 10 eggs and (**C**) = 15 eggs) on the age-stage-specific life expectancy (*e_xj_*) of the *O. strigicollis*; N1 = 1st Instar, N2 = 2nd Instar, N3 = 3rd Instar, N4 = 4th Instar, N5 = 5th Instar.

**Figure 7 insects-11-00275-f007:**
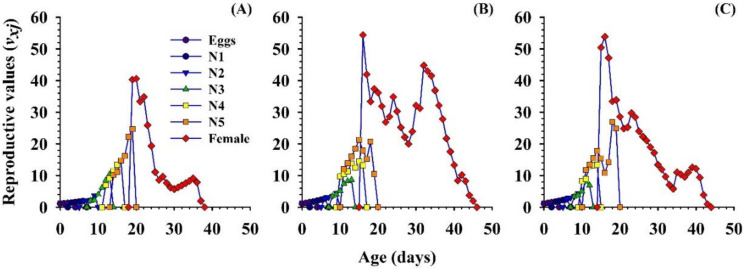
Influence of three different densities of *P. gossypiella* eggs ((**A**) = 5 eggs, (**B**) = 10 eggs and (**C**) = 15 eggs) on the age-stage reproductive value (*v_xj_*) of the *O. strigicollis*; N1 = 1st Instar, N2 = 2nd Instar, N3 = 3rd Instar, N4 = 4th Instar, N5 = 5th Instar.

**Figure 8 insects-11-00275-f008:**
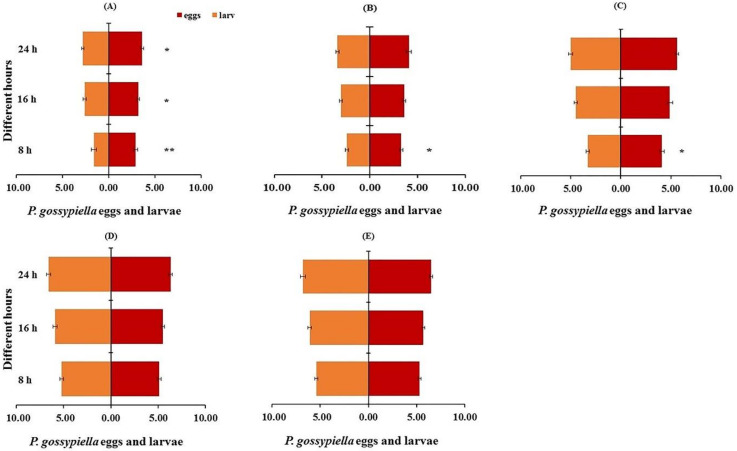
Prey preference between *P. gossypiella* eggs and first instar larvae (10 each) of the predatory stages of *O. strigicollis* during 8 h, 16 h and 24 h treatment using paired *t*-test, *n* = 10, * *p* < 0.05, ** *p* < 0.01, ((**A**) = 3rd instar, (**B**) = 4th Instar, (**C**) = 5th Instar, (**D**) = Male, (**E**) = Female).

**Table 1 insects-11-00275-t001:** Total feeding of *O. strigicollis* on Pectinophora gossypiella eggs.

Stages	Prey Densities						ANOVA		
	*n*	5 Eggs	*n*	10 Eggs	*n*	15 Eggs	*df*	*F*	*p*
1st instar nymph	27	3.33 ± 0.29 b	28	3.89 ± 0.29 a,b	29	4.41 ± 0.28 a	2, 81	3.56	0.03
2nd instar nymph	26	4.58 ± 0.38 a	28	5.04 ± 0.37 a	28	5.29 ± 0.37 a	2, 79	0.93	0.40
3rd instar nymph	26	12.54 ± 1.12 b	28	16.86 ± 1.08 a	28	19.21 ± 1.08 a	2, 79	9.43	0.00
4th instar nymph	22	10.46 ± 1.08 b	24	10.92 ± 1.03 b	23	15.65 ± 1.05 a	2, 66	7.44	0.00
5th instar nymph	18	13.56 ± 1.58 b	23	20.78 ± 1.40 a	20	21.70 ± 1.50 a	2, 58	8.35	0.00
Male	8	75.00 ± 13.46 a	8	126.50 ± 28.06 a	8	117.00 ± 30.67 a	2, 21	1.18	0.33
Female	10	67.40 ± 12.20 b	15	159.40 ± 30.11 a,b	12	188.33 ± 33.66 a	2, 34	3.16	0.05

The standard errors of the mean values of the *O. strigicollis* fed on three different densities of *P. gossypiella* eggs (5, 10, and 15) were estimated by using one-way ANOVA, LSD test, and means marked with different letters are significantly different between three treatments at the 5% significant level.

**Table 2 insects-11-00275-t002:** Effect of three prey densities on the development time of *O. strigicollis* feeding on *P. gossypiella* eggs (prey).

Stages (d)	Prey Densities						ANOVA		
	*n*	5 Eggs	*n*	10 Eggs	*n*	15 Eggs	*df*	*F*	*p*
Egg duration	60	3.23 ± 0.06 a	60	3.23 ± 0.06 a	60	3.23 ± 0.06 a	2,177	0.00	1.00
1st instar nymph	60	3.00 ± 0.00 a	60	3.00 ± 0.00 a	60	3.00 ± 0.00 a	2,177	0.00	1.00
2nd instar nymph	60	2.93 ± 0.03 a	60	2.93 ± 0.03 a	60	2.93 ± 0.03 a	2,177	0.00	1.00
3rd instar nymph	26	3.38 ± 0.12 a	34	3.00 ± 0.19 a	32	2.31 ± 0.20 b	2,89	8.70	0.00
4th instar nymph	22	2.55 ± 0.11 a	30	2.07 ± 0.14 b	28	1.93 ± 0.11 b	2,77	5.83	0.00
5th instar nymph	18	3.67 ± 0.11 a	28	3.93 ± 0.15 a	28	3.93 ± 0.15 a	2,71	0.84	0.44
Male	8	7.50 ± 1.94 a	8	11.25 ± 2.09 a	12	6.67 ± 1.25 a	2,25	2.01	0.16
Female	10	9.20 ± 1.64 a	20	11.40 ± 1.84 a	16	8.88 ± 2.13 a	2,43	0.54	0.59
Total longevity of male adult	8	26.00 ± 2.09 a	8	29.50 ± 2.49 a	12	23.67 ± 1.08 a	2,25	2.74	0.08
Total longevity of female adult	10	28.60 ± 1.73 a	20	29.20 ± 1.91 a	16	26.12 ± 2.10 a	2,43	0.70	0.50

The standard errors of the mean values of pre-adult development period and total longevity of male and females were estimated by using 200,000 bootstrap replicates. Means marked with different letters are significantly different between three treatments using the paired bootstrap test at the 5% significant level.

**Table 3 insects-11-00275-t003:** Effect of three prey densities on the biological traits of *O. strigicollis* feeding on *P. gossypiella* eggs.

Stages (d)		Prey Densities					ANOVA		
	*n*	5 Eggs	*n*	10 Eggs	*n*	15 Eggs	*df*	*F*	*p*
TPOP/TPRP ^a^ (d)	10	19.80 ± 0.13 a	18	18.00 ± 0.36 b	16	17.50 ± 0.50 b	2,43	6.81	0.00
APOP/APRP ^b^ (d)	10	0.40 ± 0.16 a	18	0.70 ± 0.17 a	16	0.25 ± 0.11 a	2,43	1.58	0.22
Oviposition (d)	10	7.20 ± 1.18 a	18	10.22 ± 1.98 a	16	7.88 ± 1.92 a	2,43	0.12	0.89
Post-Oviposition (d)	10	0.60 ± 0.40 a	18	1.40 ± 0.48 a	16	0.50 ± 0.34 a	2,43	1.39	0.26
Fecundity (eggs/female)	10	54.40 ± 9.67 a	18	90.44 ± 17.87 a	16	66.12 ± 9.38 a	2,43	0.84	0.44
Hatchability (%)	10	29.63 ± 2.73 a	18	31.46 ± 2.03 a	16	26.44 ± 2.16 a	2,41	1.45	0.25

The standard errors of the mean values of TPOP, APOP, Oviposition and Fecundity were estimated by using 200,000 bootstrap replicates. Means marked with different letters are significantly different between three treatments using the paired bootstrap test at the 5% significant level. ^a^ TPOP: Total pre-oviposition period of female, TPRP: Total pre-reproduction period of female. ^b^ APOP: Adult pre-oviposition period of female, APRP: Adult pre-reproduction period of female.

**Table 4 insects-11-00275-t004:** Effect of three prey densities on the life table parameters of *O. strigicollis* feeding on *P. gossypiella* eggs.

Population Parameters	Prey Densities		
	5 Eggs	10 Eggs	15 Eggs
Intrinsic rate of increase (*r*) (d^−1^)	0.09 ± 0.02 a	0.14 ± 0.01 a	0.14 ± 0.01 a
Finite rate of increase (*λ*) (d^−1^)	1.10 ± 0.02 a	1.15 ± 0.01 a	1.15 ± 0.02 a
Net reproductive rate (*R*_0_) (offspring)	9.07 ± 3.05 a	27.13 ± 7.47 a	17.63 ± 4.50 a
Mean generation time (*T*) (d)	23.57 ± 0.39 a	23.21 ± 1.04 a,b	20.89 ± 0.79 b
Gross reproductive rate (*GRR*)	51.03 ± 11.68 a	129.61 ± 27.26 a	90.32 ± 16.91 a

Standard errors were estimated by using 200,000 bootstrap replicates. Means marked with different letters are significantly different between three treatments using the paired bootstrap test at the 5% significant level, *n* = 60.

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
