# Peer review of "Using a Two-Sex Life Table Tool to Calculate the Fitness of Orius strigicollis as a Predator of Pectinophora gossypiella"

_insects, 2020, doi:10.3390/insects11050275_

Round 1

Reviewer 1 Report

Dear Authors

line 21-23: please make it more clear what do you report for the first time: the fitness of Orius simiis on Pectinophora gossypiella or using sex life table?

line 28: instead of "reduced generation "shorter devolpment can be? 

line 37-38: It is not always true so I suggest to eliminate and Orius is a predator for different insects belong to fifferent families

Line 45: Is it important "including China"?

Line 44-45: what do you mean with insects? I think you should rewrite 

Line 46-47: what does  "the voracity of the prey "mean?

Line 38-47: I suggets to make togheter and avoid replications

Line 56-72: to much replication, rewrite this part and avoid replications of meanings

Line 130: You mentioned 30 replicates for each experiment but it is not clear how many individuals do you use in each replicate? 

line 291: female3?

Author Response

Dear Reviewer:

On behalf of all authors, I am again thankful to you and reviewers for valuable suggestions on our manuscript # insects-768568 entitled "Using a two-sex life table tool to calculate the fitness of Orius strigicollis as a predator of Pectinophora gossypiella." Those comments are very valuable and helpful for revising and improving the manuscript, as well as the important guiding significance to our future research.

Reviewer 2 Report

General overview: too many redundant and useless particular as well as repetition (eg: the size of petri dishes). Modify accordingly.

Material and methods is often confused too useless details and substantial lack of schematic. Subsections in M&M should correspond in Results. In the present form it is extremely difficult to follow the flow of the work. Table 2 and section 3.3.1: where is the description in M&M?.

A significant fault is in statistical part of the work. Authors declare the use of ANOVA or t-test but they do not confirm if their data meet ANOVA as well as t-test requirements and they do not report results and statistic in the correct way in the text: F statistic has 2 df values!

Figures 1-3: Y axis title is misleading as data plotted are not “predation potential” but the number of killed preys

Section 3.1 is in my opinion substantially incorrect because the authors do not consider the ratio prey supplied/prey killed. This ratio does not increase at higher prey density. Authors should re-evaluate their finding /analysis taking into account this. Moreover regressions are without significance. With just 3 point to fit, it is not possible to describe trends different from linearity. Regressions do not have specific comments in the text. If 5 eggs is a limiting value why some eggs were not eaten?

How data in section 3.1 and 3.2 are related? This is not clearly explained. The total consumption by the females is not statistically different (see table 1).

Discussion and Conclusions are repetitive and verbose in some parts and general conclusion are questionable: only generation time was affected by prey availability and none of the other life parameters was significantly influenced (table 4) and only in 3rd and 4th instar the development was significantly affected by the number of supplied preys (lines 479-480)

Minor issues

Avoid repetition of “Orius similis Zheng” and respect naming conventions.

Line 22 vs line 61: do not use two different common names

Lines 29-31: check the sentence as it seems to be incomplete.

Line 116: check the sentence: efficiently does not make sense in the context.

Line 131-133: egg mortality needs validation and reporting. Why egg mortality should be density depended as written in the sentence?

Line 139: check the sentence. Why biological?

Line 167: “adults” repeated

Line 182: the difference between APOP and TPOP is not clearly stated

Lines 338-340: I suppose that this sentence should not be here.

Author Response

(The authors gave the same response as above.)

Reviewer 3 Report

The manuscript is clear and well written, the methodology is appropriate and data analysis is correct. The results are discussed and a conclusion is formulated. Therefore, in my opinion, the manuscript could be accepted in Insects. 

Author Response

(The authors gave the same response as above.)

Round 2

Reviewer 2 Report

After the proposed changes I think that the paper can be accepted